# Horizontal transfer of whole mitochondria restores tumorigenic potential in mitochondrial DNA-deficient cancer cells

Lan-Feng Dong[1]*, Jaromira Kovarova[2], Martina Bajzikova[2], Ayenachew Bezawork-Geleta[1], David Svec[2], Berwini Endaya[1], Karishma Sachaphibulkij[1], Ana R Coelho[2,3], Natasa Sebkova[2,4], Anna Ruzickova[2], An S Tan[5], Katarina Kluckova[2], Kristyna Judasova[2], Katerina Zamecnikova[2,6], Zuzana Rychtarcikova[2,7], Vinod Gopalan[1,8], Ladislav Andera[2], Margarita Sobol[9], Bing Yan[1], Bijay Pattnaik[10], Naveen Bhatraju[10], Jaroslav Truksa[2], Pavel Stopka[4], Pavel Hozak[9], Alfred K Lam[8], Radislav Sedlacek[9], Paulo J Oliveira[3], Mikael Kubista[2,11], Anurag Agrawal[10], Katerina Dvorakova-Hortova[2,4], Jakub Rohlena[2], Michael V Berridge[5]*, Jiri Neuzil[1,2]*

[1]School of Medical Science, Griffith University, Southport, Australia; [2]Institute of Biotechnology, Czech Academy of Sciences, Prague, Czech Republic; [3]CNC-Center for Neuroscience and Cell Biology, University of Coimbra, Cantanhede, Portugal; [4]Department of Zoology, Faculty of Science, Charles University, Prague, Czech Republic; [5]Malaghan Institute of Medical Research, Wellington, New Zealand; [6]Zittau/Goerlitz University of Applied Sciences, Zittau, Germany; [7]Faculty of Pharmacy, Charles University, Hradec Kralove, Czech Republic; [8]School of Medicine, Griffith University, Southport, Australia; [9]Institute of Molecular Genetics, Czech Academy of Sciences, Prague, Czech Republic; [10]CSIR Institute of Genomics and Integrative Biology, New Delhi, India; [11]TATAA Biocenter, Gothenburg, Sweden

*For correspondence: l.dong@ griffith.edu.au (L-FD); mberridge@ malaghan.org.nz (MVB); j.neuzil@ griffith.edu.au (JN)

**Competing interests:** The authors declare that no competing interests exist.

**Abstract** Recently, we showed that generation of tumours in syngeneic mice by cells devoid of mitochondrial (mt) DNA ($\rho^0$ cells) is linked to the acquisition of the host mtDNA. However, the mechanism of mtDNA movement between cells remains unresolved. To determine whether the transfer of mtDNA involves whole mitochondria, we injected B16$\rho^0$ mouse melanoma cells into syngeneic C57BL/6N$^{su9\text{-}DsRed2}$ mice that express red fluorescent protein in their mitochondria. We document that mtDNA is acquired by transfer of whole mitochondria from the host animal, leading to normalisation of mitochondrial respiration. Additionally, knockdown of key mitochondrial complex I (NDUFV1) and complex II (SDHC) subunits by shRNA in B16$\rho^0$ cells abolished or significantly retarded their ability to form tumours. Collectively, these results show that intact mitochondria with their mtDNA payload are transferred in the developing tumour, and provide functional evidence for an essential role of oxidative phosphorylation in cancer.

## Introduction

Mitochondria are vital organelles of eukaryotic cells responsible for energy production and other key biochemical functions. Many human diseases, including cancer, are characterised by mutations in mitochondrial DNA (mtDNA). This is linked to altered mitochondrial bioenergetics, such that cancer cells are more glycolytic than their non-malignant counterparts, as postulated almost a century ago by Warburg. Altered energy metabolism is now regarded as one of the hallmarks of cancer

(*Koppenol et al., 2011*; *Hanahan and Weinberg, 2011*; *Vyas et al., 2016*). In this context, the mitochondrial genome has been reported to play a role in tumorigenesis (*Lee and St John, 2016*; *Patananan et al., 2016*) and in metastatic cancer (*Ishikawa et al., 2008*; *Hayashi et al., 2016*).

Although many cancer cells are biased towards the glycolytic metabolism, they also need oxidative phosphorylation (OXPHOS) for their 'pathophysiological' requirements (*Weinberg et al., 2010*). Recent research has demonstrated a new concept of cancer metabolism, emphasising the importance of OXPHOS in the tumour environment beyond its role in bioenergetics (*Gentric et al., 2016*). A novel paradigm has emerged, according to which respiration is important for cancer cell proliferation, and also for tumour formation, progression and metastasis (*LeBleu et al., 2014*; *Viale et al., 2014*). This is probably linked to metabolic re-modelling (*Birsoy et al., 2015*; *Sullivan et al., 2015*). There are reports that some cancer cells retain OXPHOS capacity and have no obvious respiratory defects (*Frezza and Gottlieb, 2009*; *Jose et al., 2011*). Further, inhibiting glycolysis may restore higher rates of OXPHOS in neoplastic cells (*Moreno-Sánchez et al., 2007*; *Michelakis et al., 2010*). Hence, depriving cancer cells of their capacity to respire may preclude them from forming tumours.

We recently reported on the importance of respiration in tumour formation and progression (*Tan et al., 2015*). We showed that cancer cells without mtDNA ($\rho^0$ cells) form tumours after a considerable delay compared to their parental counterparts. Tumour progression was associated with mtDNA acquisition from the host, resulting in respiration recovery. While these findings point to a new phenomenon of horizontal transfer of mtDNA between mammalian cells in vivo (*Berridge et al., 2015*, *2016*), direct evidence for the role of mitochondrial respiration in tumour formation as well as understanding the mode of mtDNA acquisition is lacking.

Here we provide a link between efficient tumour formation and recovery of mitochondrial respiration, and show that mtDNA acquisition occurs via trafficking of whole mitochondria.

## Results

### Cell lines derived from tumours that formed from B16$\rho^0$ cells are homogeneous in mtDNA distribution and contain a fully assembled respirasome

We have shown that B16$\rho^0$ cells injected subcutaneously into C57BL/6J mice formed syngeneic tumours with a 2–3-week delay compared to B16 cells, and next generation sequencing (NGS) indicated the host origin of mtDNA (*Tan et al., 2015*). Since NGS would not detect heteroplasmy of less than about 3%, a much more sensitive single cell/digital droplet (sc/dd) PCR method was used in the current study to document that the homoplasmic polymorphism at the $tRNA^{Arg}$ locus of mtDNA of cell lines isolated from tumours grown subcutaneously from B16$\rho^0$ cells (B16$\rho^0$SC cells) is of the host origin. The assay is able to detect heteroplasmy down to 0.5%, demonstrating with very high confidence that the mtDNA in B16$\rho^0$SC cells is of host origin, and that original B16 polymorphism is either completely absent or present below the detection limit of 0.5% of mtDNA (*Figure 1A*; see also *Appendix 1—figure 1* for validation of sc/ddPCR).

We next analysed the properties of B16, B16$\rho^0$ and B16$\rho^0$SC cells, as well as a B16$\rho^0$CTC subline derived from circulating tumor cells and a B16$\rho^0$SCL sub-line derived from lung metastases (*Tan et al., 2015*). *Appendix 1–figure 2A* documents confocal microscopy analysis of mtDNA in mitochondria, showing that B16$\rho^0$ cells lack mtDNA, whereas mtDNA appears homogeneously distributed in mitochondria in all other sub-lines. Super-resolution stimulated emission depletion (STED) microscopy exerted similar levels and distribution of mtDNA nucleoids in B16, B16$\rho^0$SC, B16$\rho^0$CTC and B16$\rho^0$SCL cells, and no nucleoids in B16$\rho^0$ cells, also showing largely unchanged levels of Tom20 and low level of TFAM (*Figure 1B*; see also *Appendix 1—figure 2E*). Native blue gel electrophoresis (NBGE) revealed that B16$\rho^0$ cells do not contain the supercomplex/respirasome (formed by CI, CIII and CIV), but contain low amounts of sub-CV (*Figure 1C*). CII was found to be fully assembled in all sub-lines, which is reasonable considering that all four subunits in CII are encoded by nuclear DNA (nDNA) (*Figure 1C*). To test if cells replicate their mtDNA, we established the mitochondrial chromatin immunoprecipitation (mitoChIP) assay. This showed a high level of DNA polymerase-$\gamma$1 (POLG1) binding to the *D-LOOP* region of mtDNA in all cells except B16$\rho^0$ cells (*Figure 1D*). B16$\rho^0$SC, B16$\rho^0$CTC and B16$\rho^0$SCL cells showed similar respiration to B16 cells, but no respiration was observed with B16$\rho^0$ cells (*Figure 1E,F*). Accordingly, B16$\rho^0$ cells produced more

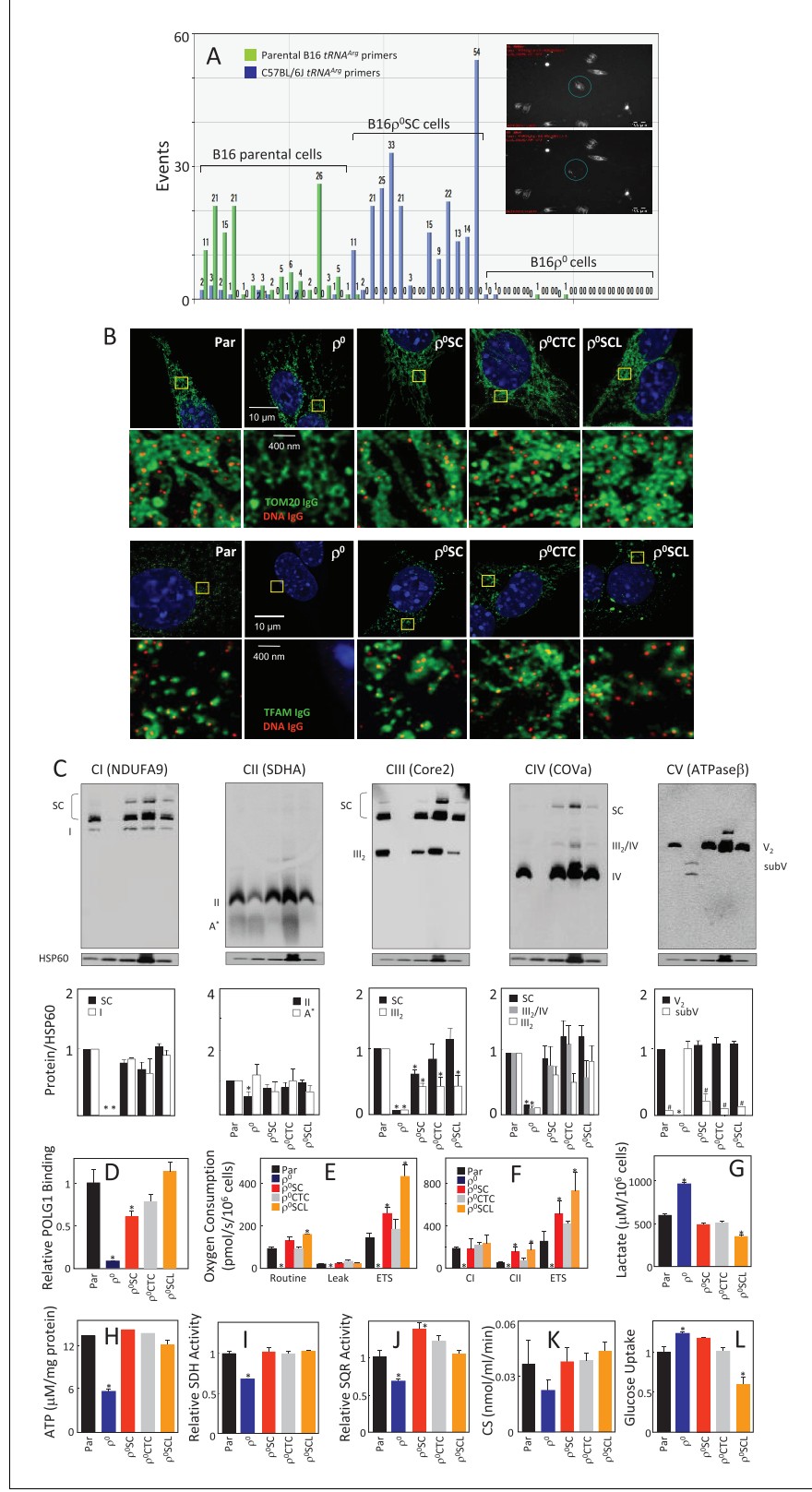

**Figure 1.** Cells derived from B16ρ° cell-grown tumours feature mtDNA with host polymorphism, and recovered mitochondrial complexes and respiration. (**A**) B16, B16ρ⁰ and B16ρ⁰SC cells were assessed by sc/dd PCR for polymorphism of the *tRNA^Arg* locus of mtDNA using specific probes (see Materials and methods). The insert shows a cell (circled) before (upper image) and after (lower image) withdrawn for analysis. (**B**) B16, B16ρ⁰, B16ρ⁰SC,

*Figure 1 continued on next page*

*Figure 1 continued*

B16ρ⁰CTC and B16ρ⁰SCL cells were immunostained for anti-DNA (red) and anti-Tom20 or anti-TFAM IgGs (green). The upper panels represent lower resolution confocal images depicting a major part of a whole cell, the lower panels represent higher magnification STED images of the region of interest indicted by the yellow box. (C) Cells as above were subjected to NBGE followed by WB using antibodies against subunits of individual complexes. Below is a densitographic evaluation of three gels derived from individual experiments with HSP60 as the internal control. The cells were assessed for binding of POLG1 to the *D-LOOP* region of mtDNA using the mitoChIP assay (D), for routine respiration (E) and for respiration via CI and CII following their permeabilisation (F). The sub-lines were next assessed for lactate generation (G), ATP level (H), SDH (I), SQR (J) and CS activities (K) as well as for glucose uptake (L). The symbol '*' indicates statistically significant differences between individual sublines and B16 cells, the symbol '#' in panel C indicates statistically significant difference between individual sublines and B16ρ⁰ cells. The nature of the individual sublines derived from B16ρ⁰ cells is as follows: B16ρ⁰SC cells, cells derived from primary tumour grown in B57BL mice grafted with B16ρ⁰ cells; B16ρ⁰CTC cells, the corresponding circulating tumour cells; B16ρ⁰SCL cells, the corresponding cells isolated from lung metastases.

lactate (*Figure 1G*) and less ATP (*Figure 1H*). B16ρ⁰ cells also had lower succinate dehydrogenase (SDH) (*Figure 1I*) and succinate quinone reductase (SQR) (*Figure 1J*) activity, as well as lower citrate synthase (CS) activity (*Figure 1K*). Finally, we observed higher glucose uptake in B16ρ⁰ and B16ρ⁰SC cells, and lower uptake in B16ρ⁰SCL cells (*Figure 1L*). Collectively, these results document that mitochondrial function is already fully restored in cells derived from the primary tumour.

We next analysed cells for their mtDNA levels and expression of selected transcripts. *Appendix 1—figure 2B* shows no mtDNA in B16ρ⁰ cells, while mtDNA was present at similar levels in other sub-lines. No mtDNA-encoded transcripts were present in B16ρ⁰ cells. Their levels were low in B16ρ⁰SC cells, while higher levels were seen for most transcripts in B16ρ⁰CTC and in B16ρ⁰SCL cells. Transcripts of the assembly factor SCAFI were present at similar levels in all sub-lines, but TFAM transcripts were low in B16ρ⁰ cells relative to the other cells. Transcripts of nDNA genes coding for subunits of respiratory complexes were found to be present in all cells, with some being lower in B16ρ⁰ cells. WB revealed that most proteins investigated were relatively abundant in all sub-lines with LC3AII levels lower in B16ρ⁰ cells, indicating stalled autophagy (*Appendix 1—figure 2C*). Interestingly, although present in B16ρ⁰ cells, many nDNA-encoded mitochondrial proteins, including subunits of RCs, were unstable in these cells, as evidenced using cycloheximide treatment (*Appendix 1—figure 2D*). Exceptions were SDHA and ATPβ; a plausible reason is the absence of binding partners encoded by mtDNA, which could render the 'unassembled nDNA-encoded' subunits unstable. In summary, these results indicate that, in contrast to 4T1 cells (*Tan et al., 2015*), the respiratory function of B16 sublines is already fully recovered at the primary tumour stage.

## Recovery of B16ρ⁰ cell respiration fully restores their propensity to form tumours

Given the rapid recovery of respiratory function in the B16 model, we next tested tumour-forming capacity of B16 and B16ρ⁰ cells, and of the sub-lines B16ρ⁰SC, B16ρ⁰CTC and B16ρ⁰SCL, derived from various tumour stages as described above. We found that all sub-lines formed tumours without delay except for B16ρ⁰ cells with >2 week delay (*Figure 2A*). Sectioning of tumours derived from all five sub-lines revealed melanomas with necrotic cells away from blood vessels (*Figure 2B*). Tumours derived from the sub-lines were assessed by NBGE, identifying full assembly of all complexes as well as the respirasome, with somewhat higher levels of the respirasome in tumours derived from B16ρ⁰ cells (*Figure 2C*). WB revealed similar levels of expression of most proteins tested in the five types of tumours (*Figure 2D*). No obvious differences in the expression of genes coding for transcripts of individual subunits of mitochondrial complexes encoded by mtDNA and nDNA were observed (*Figure 2E*). Finally, we tested respiration of tumour tissues derived from the sub-lines. *Figure 2F* reveals increased CI-dependent respiration and the maximum electron transfer capacity (ETS) in tumours derived from B16ρ⁰ and from B16ρ⁰SCL cells, and no change in CII-dependent respiration. Normal liver tissue from the same animals used as internal control showed no changes in respiration. The restoration of the tumorigenic potential fully correlates with the observed recovery of respiration.

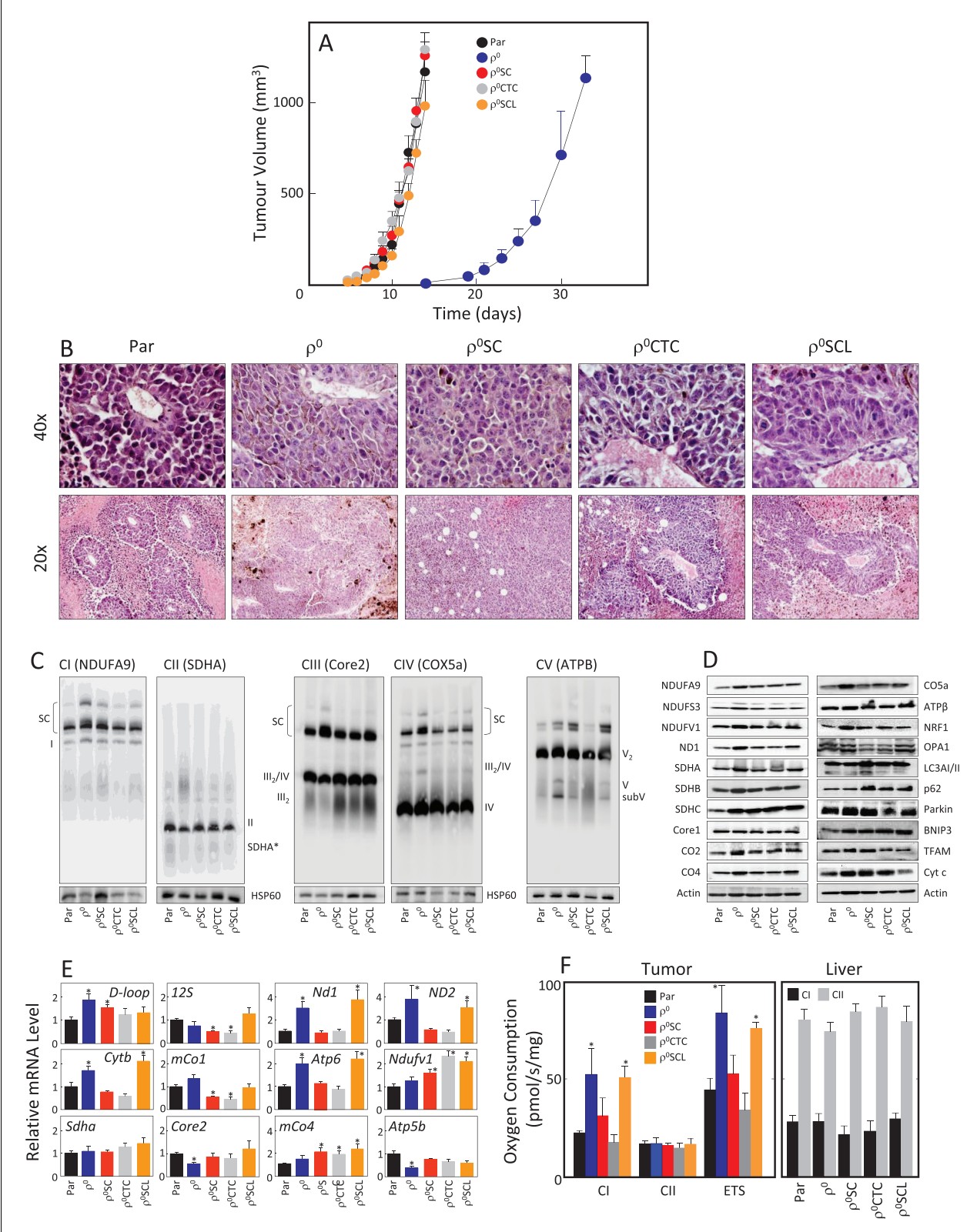

**Figure 2.** B16ρ⁰ cells form tumours with a delay and with fully assembled respirasome. (**A**) B16, B16ρ⁰, B16ρ⁰SC, B16ρ⁰CTC and B16ρ⁰SCL cells were grafted in C57BL/6J mice ($10^6$ cells per animal; 6 mice per group), and tumor growth was evaluated using USI. (**B**) Tumours derived from individual sublines were fixed and sectioned, and inspected following H and E staining. (**C**) Individual tumours were subjected to NBGE followed by WB to visualize mitochondrial SCs and RCs. SDS-PAGE followed by WB with antibodies to subunits of mitochondrial RCs and other proteins was used to assess their

*Figure 2 continued on next page*

*Figure 2 continued*
levels (D), qPCR was used to assess the levels of representative mtDNA- and nDNA-coded mRNAs (E). (F) Tumor (left) and liver tissues (right) from mice grafted with individual B16 sub-lines were assessed for CI- and CII-dependent respiration, and for maximal uncoupled respiration (ETS, tumour only). The symbol '*' indicates statistically significant differences between tumours derived from individual sub-lines and tumours derived from B16 cells.

## Suppression of respiration interferes with efficient tumour formation

We have shown for B16 cells (*Figures 1* and *2*, *Appendix 1—figure 2*) and previously for 4T1 cells (*Tan et al., 2015*) that tumour formation from their respective $\rho^0$ variants correlates with the acquisition of mtDNA and recovery of respiration. However, direct evidence for the requirement of respiration for tumour formation has been missing. We therefore prepared B16 and B16$\rho^0$ cells with suppressed levels of NDUFV1 (the catalytic subunit of CI) or SDHC (ubiquinone-binding CII sub-unit essential for its SQR activity) by RNA interference (RNAi) using two different shRNAs for each protein. *Figure 3A* shows that NDUFV1 shRNA#2 and SDHC shRNA#2 were rather efficient in knocking down the respective proteins. *Figure 3A* also reveals that NDUFV1 knock-down (KD) cells exhibit lower level of CI subunits (NDUFV1 and NDUFS3) while CII subunits (SDHA and SDHC) were unaffected. Conversely, SDHC KD cells showed low levels of CII subunits and also decreased levels of CI subunits. We used cells stably transfected with NDUFV1 shRNA#2 or SDHC shRNA#2 in subsequent experiments. Both NDUFV1 KD and SDHC KD cells proliferated at a slower rate than parental cells (*Figure 3B*). NDUFV1 KD as well as SDHC KD cells showed lower routine respiration and lower ETS (*Figure 3C*). NDUFV1 KD cells respired less via CI, while CII-dependent respiration was largely unaffected; SDHC KD cells not only showed much lower CII-dependent respiration, but also significantly suppressed CI-dependent respiration (*Figure 3D*).

We next grafted B16, B16$\rho^0$ and B16 cells, as well as their derived NDUFV1 KD and SDHC KD cells into C57BL/6J mice. *Figure 3E* shows that tumours started to grow from B16$\rho^0$ cells shortly after day 20 post-grafting of B16$\rho^0$ cells in all mice. 4 out of 6 mice grafted with B16$\rho^0$ SDHC KD cells formed tumours with delays of 15 to 40 days compared to B16$\rho^0$ cells. Only one out of 5 mice of the B16$\rho^0$ NDUFV1 KD group formed a tumour, with a lag of about 40 days compared to B16$\rho^0$ cells. A similar pattern was observed for B16 cells and their NDUFV1 KD and SDHC KD variants, though the lag for these sub-lines was considerably shorter (*Figure 3F*). On pathological examination of the different tumours, relatively subtle differences in morphology were observed (*Figure 3G*). B16 melanoma cells form tumours in control mice (NS) showed prominent nuclear pleomorphism. They appeared more aggressive than cells from $\rho^0$ tumours. Tumours derived from both B16 NDUFV1 KD and B16 SDHC KD cells showed better histological differentiation associated with eosinophilic cytoplasm, vesicular nuclei and frequent deposits of melanin pigments, indicating the KD cells are less aggressive than their parental counterpart. There were no obvious differences in morphology of the 3 types of $\rho^0$ melanoma cells. Both B16$\rho^0$ NDUFV1 KD and B16$\rho^0$ SDHC KD tumours showed focal areas of nuclear pleomorphism and frequent mitotic features.

These data clearly point to respiration recovery as essential for driving efficient tumour formation, since suppression of respiration completely deregulated this process. Interestingly, while with CII suppression the majority of mice formed tumours (albeit with additional lag time), most mice grafted with CI-compromised cells failed to form tumours within 100 days. In summary, we show that respiration is important for efficient tumour formation, which is consistent with recent reports (*Weinberg et al., 2010*; *Birsoy et al., 2015*; *Sullivan et al., 2015*).

## B16$\rho^0$ cells acquire mtDNA via transfer of whole mitochondria from the host

We have previously documented the host origin of mtDNA in cancer cells isolated from primary tumours derived from 4T1$\rho^0$ and B16$\rho^0$ cells based on NGS analysis (*Tan et al., 2015*), which we confirmed here for B16$\rho^0$SC cells using the more sensitive sc/dd PCR assay (*Figure 1A*). The only plausible explanation for this phenomenon is the transfer of mtDNA from host cells to tumour cells with compromised mtDNA. The question of how mtDNA moves between cells has not been addressed. An attractive scenario is that whole mitochondria with their payload of mtDNA are transferred, but alternative explanations such as cell fusion have also been suggested (*Vitale et al.,*

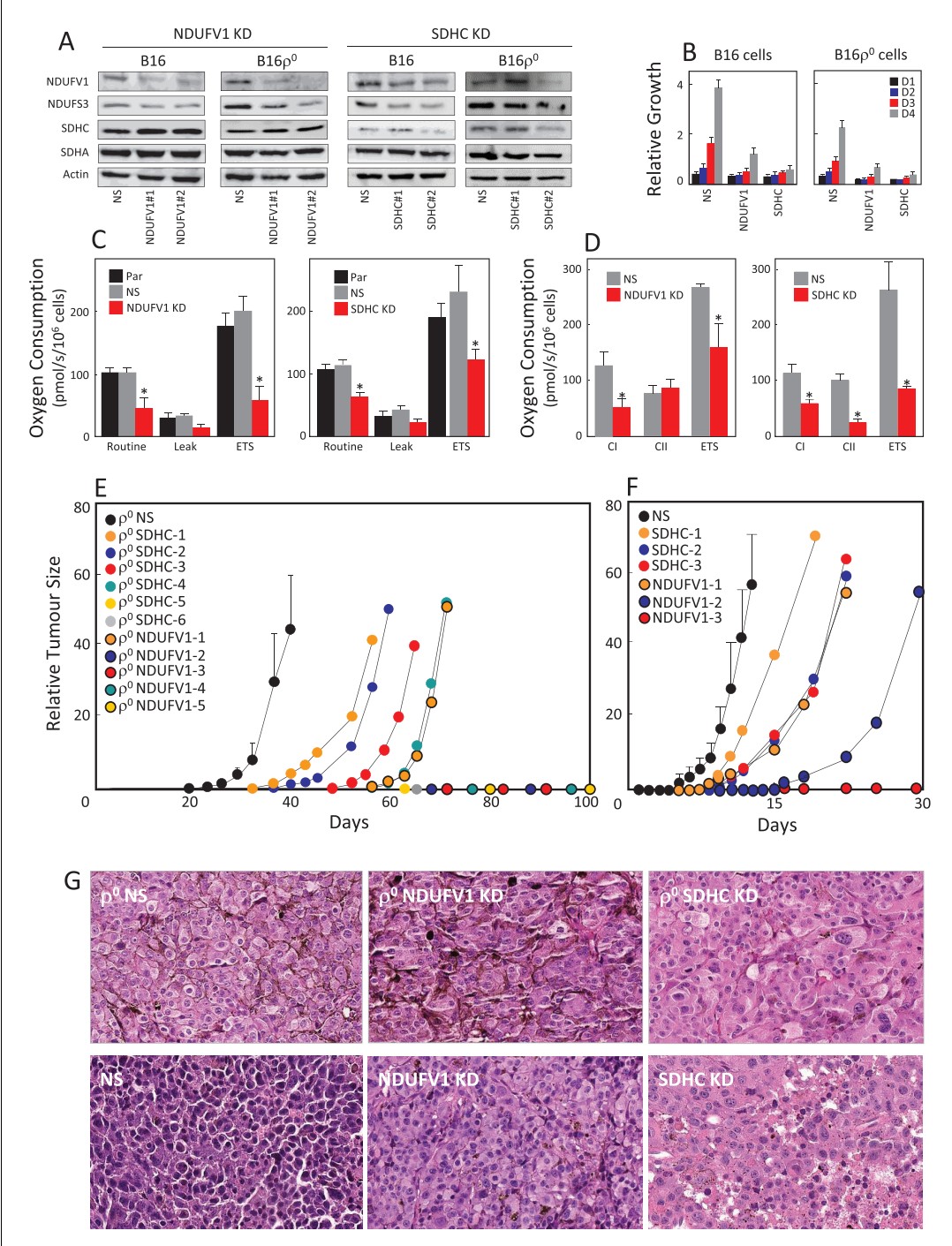

**Figure 3.** Suppression of respiration deregulates tumour growth. (**A**) B16 and B16$\rho^0$ cells were stably knocked down for either NDUFV1 or SDHC, or transfected with non-silencing (NS) shRNA, and the expression of NDUFS3, NDUFV1, SDHA and SDHC was assessed by WB. The sub-lines were next evaluated for proliferation on days 1 (D1), D2, D3 and D4 (**B**), for routine, leak and ETS respiration (**C**), and for respiration via CI and CII (**D**). Balb/-c nude mice were injected s.c. with $5 \times 10^6$ B16$\rho^0$ cells as well as cells with NDUFV1 KD or SDHC KD cells (**E**), or with B16 cells and the derived NDUFV1 KD or SDHC KD cells (**F**), with six mice per group except for the NDUFV1 group with five animals. Individual mice were assessed for tumor volume using USI. Tumours derived from B16$\rho^0$ and B16 cells were averaged and plotted as mean values, while tumours derived from knock-down cells were plotted individually. Circles on the X-axis represent individual mice in which tumours did not form within the duration of the experiment. (**G**) Mice with tumours derived from cell lines as shown in the Figure were sacrificed, and tumours fixed, sectioned and stained with H and E. The symbol '*' indicates statistically significant differences between B16 cells and NDUFV1 KD or SDHC KD cells.

*2011*; *Maiuri and Kroemer, 2015*). Mesenchymal stem cells (MSCs) have been proposed as donors of mitochondria due to their high levels of Miro-1, which is the adaptor protein responsible for mitochondrial association with the microtubule mobility complex (*Ahmad et al., 2014*; *Mills et al., 2016*). We performed co-culture of MSCs isolated from C57BL/6J mouse bone marrow with either B16 or B16$\rho^0$ cells. Prior to co-culture, the mitochondria of the potential donor (MSC or B16) cells were labelled with mitochondrial dark red fluorophore (MitoDR). Carboxyfluorescein succinimidyl ester (CFSE) that binds covalently to amino acid residues inside cells was used for staining the potential recipient (B16 or B16$\rho^0$) cells. Following co-culture, cells were assessed by confocal microscopy to determine the extent of mitochondrial transfer to recipient cells. Representative images and quantitative evaluation (*Figure 4A*, *Appendix 1—figure 3A,B*) reveal accumulation of MitoDR stained mitochondria in CFSE stained $\rho^0$ cells, as a result of their transfer from MitoDR stained MSCs. In similar experiments, pairs of either CFSE$^+$B16 / MitoDR$^+$MSC or CFSE$^+$B16$\rho^0$/MitoDR$^+$B16 cells did not show any evidence of MitoDR stained mitochondria in CFSE positive cells, thereby excluding dye diffusion. Collectively, this indicates that transfer of mitochondria between cells in a regulated process.

We next conducted a co-culture experiment, in which MSCs were isolated from transgenic C57BL/6N$^{su9-DsRed2}$ mice with red fluorescent mitochondria in somatic cells. These cells were cultured with $\rho^0$ cells transfected with a plasmid coding for nuclear targeted blue fluorescence protein (nBFP) and plasma membrane green fluorescence protein (pmGFP). *Appendix 1—figure 3C* documents transfer of DsRed mitochondria from MSCs into the recipient $\rho^0$ cells via an intracellular bridge.

To investigate the origin of mtDNA and the manner of its transfer between host and cancer cells in vivo, we used transgenic C57BL/6N$^{su9-DsRed2}$ mice with red fluorescent mitochondria in somatic cells. B16$\rho^0$ cells transfected with a plasmid coding for nuclear-targeted blue fluorescent protein (nBFP) were injected subcutaneously into C57BL/6N$^{su9-DsRed2}$ mice. Several days later, mice were sacrificed, the pre-tumour lesion excised, and single cell suspension sorted for double-positive (DP) cells with both red and blue fluorescence that on average were found with the frequency of $0.23 \pm 0.18$ in the BFP-positive population. Immediately after sorting, B16$\rho^0$ DP cells were plated and inspected by confocal microscopy within about 12 hr before red fluorescence in mitochondria has been lost. *Figure 4B* shows an image of a B16$\rho^0$ DP cell prepared from a day 11 pre-tumour lesion, identifying mouse stromal cells as a source of mitochondria that moved into a grafted B16$\rho^0$ cell. The sorted DP cells were established as a sub-line. B16, B16$\rho^0$, and B16$\rho^0$ DP cells were subjected to transmission electron microscopy and STED microscopy to show the presence of fully formed mitochondrial cristae (*Figure 4C*) and mtDNA nucleoids in DP cells (*Figure 4D*). DP cells also showed strong binding of POLG1 to the *D-LOOP* region of mtDNA (*Figure 4E*) as well as recovery of respiration (*Figure 4F*), and showed a propensity to form tumours without a lag phase (*Figure 4G*). These results document that mtDNA is transferred from stromal cells to B16$\rho^0$ cells within intact mitochondria, resulting in the restoration of respiration and in efficient tumour formation.

## Discussion

Horizontal gene transfer is a process that until recently had not been known in mammals (*Keeling and Palmer, 2008*), but has been described for lower eukaryotes (*Gladyshev et al., 2008*), affecting their phenotype (*Boschetti et al., 2012*). In mammals, the mitochondrial gene transfer has been inferred in a 10,000 year-old canine transmissible venereal tumour (CTVT) (*Rebbeck et al., 2011*; *Murchison et al., 2014*; *Strakova and Murchison, 2015*). These predictions have now been confirmed, and a recent study showed at least 5 mtDNA transfers in CTVT within the last 1–2 thousand years (*Strakova et al., 2016*). Mitochondrial transfer was reported in vitro with functional consequences (*Spees et al., 2006*; *Wang and Gerdes, 2015*), as well as in mice with endogenously injected MSCs (*Islam et al., 2012*; *Ahmad et al., 2014*), but these results could also be explained by the association of membrane-bound particles or exosomes containing mitochondria with the damaged cells rather than functional mitochondria proliferating/dividing inside cells. A recent publication from our laboratory showed, for the first time, horizontal transfer of mitochondrial genes between mammalian cells in vivo, based on acquisition of host mtDNA by $\rho^0$ tumour cells and on the presence of host mtDNA markers in cell lines derived from these tumours (*Tan et al., 2015*). Several

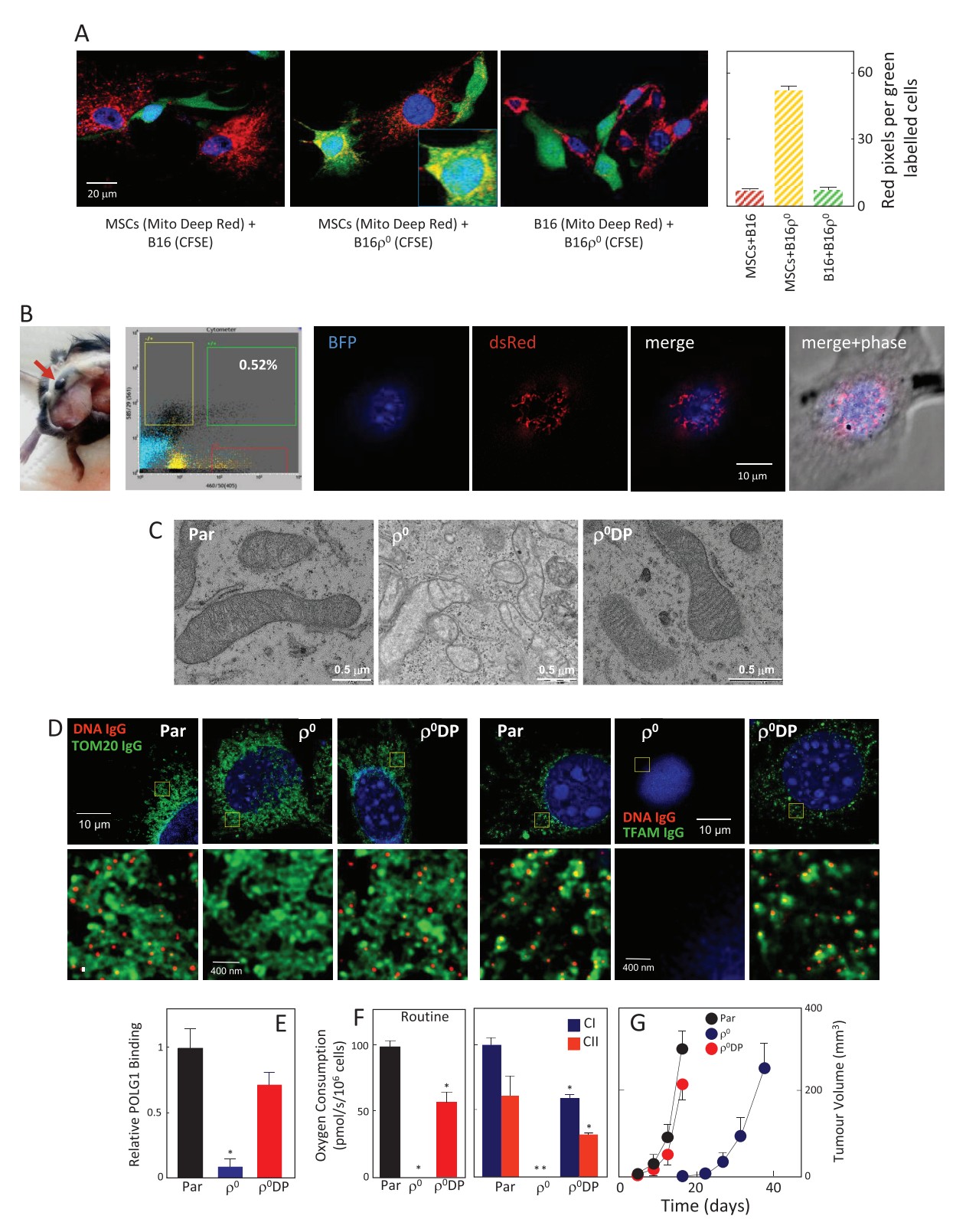

**Figure 4.** mtDNA transfers from host cells to B16ρ0 cells in whole mitochondria. (**A**) MSCs prepared from C57BL/6J mice were labelled with Mito Deep Red (MitoDR) targeted to mitochondria, B16 or B16ρ0 cells were labelled with CFSE. On the left, confocal micrographs are shown for MSCs labelled with MitoDR co-cultured with B16 or B16ρ0 cells, or B16 cells labelled with MitoDR co-cultured with B16ρ0 cells labelled with CFSE for 24 hr. On the right, evaluation of confocal microscopy is shown. (**B**) C57BL/6N[su9DsRed2] mice with red fluorescent mitochondria were grafted subcutaneously with 10[6]

*Figure 4 continued on next page*

*Figure 4 continued*

B16$\rho^0$ cells stably transfected with nBFP. After 11 days, a mouse was sacrificed and the pre-tumour lesion excised and digested into the single cell population, which was sorted for double positive (DP) (red and blue fluorescent) cells. The cells were then inspected by confocal microscopy for blue nuclei and red mitochondria. The image shows maximum intensity Z-projection of a representative DP cell. (C) Parental B16, $\rho^0$ and $\rho^0$DP cells were evaluated for mitochondrial morphology using transmission electron microscopy. (D) Parental cells and their $\rho^0$ and DP counterparts were imunnostained for DNA, TFAM and Tom20, and inspected by STED microscopy for mitochondrial nucleoids. The upper panels show a confocal image of a major part of a whole cell, the lower images depict higher magnification of the regions of interest indicated above by the yellow box obtained by STED. (E) Parental cells and their $\rho^0$ and DP counterparts were assessed for binding of POLG1 to the mtDNA *D-LOOP* region using mitoChIP. (F) Parental cells and their $\rho^0$ and DP counterparts were evaluated for routine respiration or respiration via CI and CII. (G) B16 cells and their $\rho^0$ and DP counterparts were grafted s.c. in C57BL/6J mice at $10^6$ per animal and tumour growth evaluated by USI. The symbol '*' indicates statistically significant differences between individual B16$\rho^0$ or B16$\rho^0$ DP cells and parental B16 cells.

subsequently published papers also provide evidence for horizontal transfer of mitochondria under (patho)physiological conditions (*Lei and Spradling, 2016*; *Hayakawa et al., 2016*; ,*Osswald et al., 2015*; *Moschoi et al., 2016*), pointing to OXPHOS as a factor complicating cancer therapy (*Osswald et al., 2015*; *Moschoi et al., 2016*, *Matassa et al., 2016*). Hence, our data, along with those of other groups, point to mitochondrial transfer as a highly dynamic field of research, with important implications for the conceptual understanding of cancer.

Our recent findings (*Tan et al., 2015*) provoke a number of questions, such as whether respiration is essential for tumour formation, and what is the mode of mtDNA acquisition. To address these queries, B16 metastatic melanoma cells without mtDNA and with compromised respiratory function were used. We show that B16$\rho^0$ cells do not form tumours unless they acquire mtDNA and that severe suppression of either CI- or CII-dependent respiration leads to impaired ability to form tumours, directly linking mitochondrial respiration and tumor growth. The reason(s) for the apparently greater importance of CI-dependent respiration in tumour formation (*Figure 3E,F*) remain to be explored, particularly when the role of SDH in metabolic re-modelling is taken into consideration (*Cardaci et al., 2015*; *Lussey-Lepoutre et al., 2015*). From a mechanistic point of view, CI is essential for respirasome assembly (*Moreno-Lastres et al., 2012*) that is needed for CI-dependent respiration (*Acín-Pérez et al., 2008*; *Lapuente-Brun et al., 2013*; *Tan et al., 2015*). Another indication of the importance of CI is the finding that tumours derived from B16$\rho^0$ cells feature much higher CI-dependent respiration (*Figure 2F*).

We document here that the primary tumour-derived B16$\rho^0$SC cells feature a fully assembled respirasome and complete recovery of respiration, while corresponding 4T1$\rho^0$SC cells show only partial (20–25%) recovery of respiration (*Tan et al., 2015*). Consistent with this, 4T1$\rho^0$SC cells formed tumours with a longer delay than parental 4T1 cells, while no delay was observed for B16$\rho^0$SC cells. This is likely related to the higher requirement of B16$\rho^0$ cells for respiration recovery to form tumours. This 'threshold', apparently lower for 4T1$\rho^0$ cells, may be due to higher routine respiration of B16 cells of some 100 pmol $O_2$/s/$10^6$ cells (*Figure 1E*) when compared to less than 20 pmol $O_2$/s/$10^6$ cells for 4T1 cells (*Tan et al., 2015*). Additionally, the different genetic changes in B16 melanoma and 4T1 breast carcinoma cells could underpin altered respiration recovery. Whether there is a link to a requirement for 'threshold' recovery of respiration for cancer cells to initiate tumour formation has yet to be determined. We propose the term 'OXPHOS addiction' to describe the requirement for mitochondrial respiration across the landscape of tumours, and will investigate this in more detail in future research.

A key question has been the mode of movement of mtDNA between cells. Both selective transfer of mitochondria and cell fusion have been proposed in the past (*Tan et al., 2015*; *Vitale et al., 2011*). Using C57BL/6N$^{su9-DsRed2}$ mice with red fluorescent mitochondria in somatic cells, we now provide evidence for acquisition of mtDNA by the trafficking of whole mitochondria from host donor cells to $\rho^0$ cells both in vivo (*Figure 4B*), resulting in long-lasting respiration recovery and, consequently, efficient tumour formation (*Figure 4E–G*). The transient nature of dsRed expression in recipient B16$\rho^0$ cell mitochondria in the in vivo model sheds additional light on the mechanism of mitochondrial transfer. When dsRed-containing mitochondria are selectively transferred from the host into the recipient cells, dsRed cannot be replenished in the donated mitochondria by de novo synthesis (not being encoded in the recipient's nuclear genome), and the red fluorescence in

donated mitochondria is quickly lost. On the other hand, if cell fusion were responsible for the observed acquisition of mitochondria from the donor cells (*Maiuri and Kroemer, 2015*), the donor's nuclear material would also be transferred, and mitochondrial dsRed expression would be maintained. As this is not the case, the selective transfer of whole, intact mitochondria remains the only possible explanation of our experimental data. The next step will be to investigate the mechanism of mitochondrial trafficking between cells in vivo, with tunnelling nanotubes being a plausible mode of intercellular transfer of the organelles (*Rustom et al., 2004*; *Rogers and Bhattacharya, 2014*; *Rustom, 2016*).

We conclude that recovery of respiration in tumor cells with damaged mtDNA is essential for efficient tumour formation and that this is accomplished by the intercellular transfer of whole mitochondria. Our findings are consistent with the emerging notion of the essential role of respiration in cancer cell proliferation and tumor progression (*LeBleu et al., 2014*; *Viale et al., 2014*; *Birsoy et al., 2015*; *Sullivan et al., 2015*; *Cardaci et al., 2015*; *Lussey-Lepoutre et al., 2015*; *Berridge et al., 2015*; *Viale et al., 2015*), and of a role for mitochondrial transfer in maintaining the bioenergetics balance (*Sinha et al., 2016*; *Wu et al., 2016*). From translational angle, recent studies on horizontal mitochondrial transfer indicate two tantalising, novel approaches to cancer therapy: targeting mitochondrial respiration and blocking transfer of mitochondria from stromal cells to cancer cells. While we have started exploring the first approach (*Dong et al., 2011*; *Boukalova et al., 2016*; *Kluckova et al., 2015*; *Rohlenova et al., 2017*), the other approach remains untested, with recent papers (*Osswald et al., 2015*; *Moschoi et al., 2016*) pointing to its plausibility. Finally, to the best of our knowledge, this paper is the first report to show lasting functional consequences of a well-documented mitochondrial transfer event.

# Materials and methods

## Cell culture

Cell lines were prepared and maintained as described (*Tan et al., 2015*). C57BL/6J mice were used as host animals for grafting B16 sub-lines as indicated (*Tan et al., 2015*). B16 lines formed syngeneic tumours in C57BL/6J mice indicating their authenticity. Parental and $\rho^0$ cells with BFP nuclei were prepared by stable transfection with the *pTagBFP-H2B* plasmid (Evrogen) followed by clonal selection. RNAi was used to knock down NDUFV1 or SDHC subunits, using two different shRNAs (OriGene) for each protein. In brief, cells were transfected with shRNA using a standard protocol and inspected by WB for protein levels. Cells with more efficient knock-down of NDUFV1 or SDHC were used in further experiments.

## Isolation of mesenchymal stem cells and their co-culture with cancer cells

Mouse MSCs were isolated from C57BL/6J mice as described earlier (*Ahmad et al., 2014*). The primary cells were plated at the density of $10^6$ cells/ml in T25 culture flasks and experiments were performed after the fourth passage. For co-culture experiments, MSCs were stained with Mito Deep Red (Ex/Em, 644/665 nm; Invitrogen) and B16$\rho^0$ cells with the CFSE dye (Ex/Em, 492/517 nm; Invitrogen) for 15 min. The cells were then co-cultured for 24 hr and evaluated by flow cytometry (FACS Calibur) and confocal microscopy (63 x; Leica SP8). Mitochondrial transfer was primarily ascertained by a fraction of CFSE-positive cells that were also MitoDeepRed-positive. The quantitative mitochondrial transfer was primarily ascertained by pixel counts on Z-stack confocal images i.e. red pixels in cells with green background. Geometric mean intensity of MitoDeerRed fluorescence in double positive cells was additionally calculated, as a confirmatory measurement of the degree of mitochondrial transfer. Other co-culture combinations (MSCs with B16 cells, B16 cells with B16$\rho^0$ cells) were performed in a similar fashion. Additionally, $\rho^0$ cells were transfected with a plasmid coding for nBFP and pmGFP followed by clonal selection. Transfected $\rho^0$ cells were seeded with MSCs prepared from C57BL/6N$^{su9DsRed2}$ mice in glass-bottom dishes (In Vitro Scientific) at 1:1 ratio and co-cultured for 24 hr. Live cells were inspected using the inverted fluorescence microscope Delta Vision Core with laser photo-manipulation. The acquired images were deconvolved by Huygens Professional software (Scientific Volume Imaging) and processed by FiJi ImageJ software.

## Animals

C57BL/6J mice were used for most of the experiments. They were purchased from the Animal Resources Centre or produced by the animal breeding facilities of the Institute of Biotechnology and Malaghan Institute. In all cases, the mice were grafted subcutaneously with various cell lines at $5 \times 10^5$ cells per animal. Tumours were monitored by ultrasound imaging (USI) using the Vevo770 system (VisualSonics, Toronto, Canada). Transgenic mice expressing red fluorescent protein in somatic cell mitochondria (the *CAG/su9-DsRed2* transgene) were generated in the Transgenic Unit of the Czech Centre for Phenogenomics, Institute of Molecular Genetics, Prague, Czech Republic, using a pronuclear injection from the construct provided by Prof. Masaru Okabe (Osaka University, Japan) (*Hasuwa et al., 2010*) and C57BL/6N mice. The stable colony of transgenic mice was housed in the animal facility of the Faculty of Science, Charles University, Prague, Czech Republic, and food and water were supplied ad libitum. The mice used for the grafting experiments were healthy 10 weeks old animals with no sign of stress or discomfort. All animal procedures and experimental protocols were approved by the Animal Welfare Committee of the Czech Academy of Sciences (Animal Ethics Number 18/2015).

## Single cell-digital droplet PCR

Details of the methodology are in the Supplemental information.

## Microscopic and flow cytometric cell evaluation, cell sorting and STED microscopy

Details of the methodology are in the Supplemental information.

## Mitochondrial biochemistry assays, gene expression analysis and respiration assays

Details of the methodology are in the Supplemental information.

## Statistical analysis

Unless stated otherwise, data are mean values ± S.D. of at least three independent experiments. In mouse experiments, groups of 6 animals were used, unless stated otherwise. The two-tailed unpaired Student's t test was used to assess statistical significance with $p<0.05$ being regarded as significant. Images are representative of three independent experiments.

## Acknowledgements

We thank Prof. Okabe for providing the *CAG/su9-DsRed2* plasmid, and the Transgenic Unit, Institute of Molecular Genetics, CAS, Prague, Czech Republic, for production of transgenic mouse founders. We also thank the Light Microscopy Core Facility of the Institute of Molecular Genetics, Czech Academy of Sciences, Prague, Czech Republic (supported by grants LM201504, CZ.2.16/3.1.00/21547 and LO1419) for their help with confocal/super-resolution microscopy. The work was supported in part by the Australian Research Council grant DP150102820 and grants from the Czech Science Foundation 17-0192J, 16-12719S and 15-02203S to JN, 16-22823S and 17-20904S to JR, 14-05547S to KD-H., and 16-12816S to JT. RS was supported by a grant from the Czech Academy of Sciences (RVO 68378050) and by the MEYS, LM2011032 (Czech Centre for Phenogenomics) grant. AA, NB and BP were supported by CSIR (India) grant MLP5502 and the Wellcome Trust DBT India Alliance Senior Fellowship (AA). MVB and AST were supported by the Cancer Society of New Zealand, Genesis Oncology Trust and the Malaghan Institute of Medical Research. ARC was supported by PhD scholarship from the Foundation for Science and Technology (SFRH/BD/103399/2014). The project was supported in part by TACR (TE01020118); the electron microscopy data presented in this paper were produced at the Microscopy Centre - Electron Microscopy Core Facility, IMG ASCR, Prague, Czech Republic, supported by MEYS CR (LM2015062 Czech-BioImaging). The work was also supported by the Ministry of Education, Youth and Sports of CR within the LQ1604 National Sustainability Program II (Project BIOCEV-FAR) and by the BIOCEV European Regional Development Fund CZ.1.05/1.1.00/02.0109 and by the institutional support of the Institute of Biotechnology RVO: 86652036.

## Additional information

### Funding

| Funder | Grant reference number | Author |
|---|---|---|
| Fundação para a Ciência e a Tecnologia | SFRH/BD/103399/2014 | Ana R Coelho |
| Cancer Society of New Zealand | | An S Tan<br>Michael V Berridge |
| Genesis Oncology Trust | | An S Tan<br>Michael V Berridge |
| Malaghan Institute of Medical Research | | An S Tan<br>Michael V Berridge |
| Council for Scientific and Industrial Research | MLP5502 | Anurag Agrawal<br>Naveen Bhatraju<br>Bijay Pattnaik |
| Wellcome | DBT India Alliance Senior Fellowship | Bijay Pattnaik<br>Naveen Bhatraju<br>Anurag Agrawal |
| Czech Science Foundation | 16-12816S | Jaroslav Truksa |
| Ministerstvo Školství, Mládeže a Tělovýchovy | LM2011032 | Radislav Sedlacek |
| Akademie věd České republiky | RVO 68378050 | Radislav Sedlacek |
| Czech Science Foundation | 14-05547S | Katerina Dvorakova-Hortova |
| Czech Science Foundation | 16-22823S | Jakub Rohlena |
| Czech Science Foundation | 17–20904S | Jakub Rohlena |
| Czech Science Foundation | 16–12816S | Jakub Rohlena |
| Australian Research Council | DP150102820 | Jiri Neuzil |
| Czech Science Foundation | 17–0192J | Jiri Neuzil |
| Czech Science Foundation | 16-12719S | Jiri Neuzil |
| Czech Science Foundation | 15-02203S | Jiri Neuzil |
| BIOCEV European Regional Development | | Jiri Neuzil |

The funders had no role in study design, data collection and interpretation, or the decision to submit the work for publication.

### Author contributions

L-FD, Conceptualization, Supervision, Investigation, Writing—original draft, Writing—review and editing; JK, Conceptualization, Investigation, Methodology, Writing—review and editing; MB, Investigation, Methodology, Writing—review and editing; AB-G, Conceptualization, Data curation, Investigation, Methodology; DS, KS, Data curation, Investigation, Methodology; BE, ARC, AR, KJ, KZ, ZR, LA, MS, BY, BP, Investigation, Methodology; NS, AST, VG, Resources, Investigation, Methodology; KK, Methodology, Data acquisition; NB, Data curation, Supervision, Investigation, Methodology; JT, PS, Resources, Supervision, Investigation, Methodology; PH, Supervision, Methodology; AKL, Resources, Supervision, Investigation; RS, Supervision, Investigation, Methodology; PJO, Conceptualization, Supervision, Methodology; MK, Conceptualization, Supervision, Investigation; AA, Conceptualization, Supervision, Investigation, Methodology; KD-H, Conceptualization, Supervision, Funding acquisition, Investigation, Writing—original draft, Writing—review and editing; JR, Conceptualization, Resources, Writing—original draft, Writing—review and editing; MVB, Conceptualization, Supervision, Funding acquisition, Methodology, Writing—original draft, Project administration, Writing—review and editing; JN, methodology, investigation

Author ORCIDs

Jiri Neuzil, http://orcid.org/0000-0002-2478-2460

Ethics

Animal experimentation: This study was performed in strict accordance with the recommendations in the Guide for the Care and Use of Laboratory Animals of the Czech Republic All animal procedures and experimental protocols were approved by the Local Ethics Committee (Animal Ethics Number 18/2015).

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

## Appendix 1

### Single cell-digital droplet PCR

Single cell samples were obtained by automated cell withdrawal using the CellCelector (ALS). Approximately 10,000 cells were seeded in a 6-well culture plate, nuclei were stained with DAPI and automated image analysis was performed to identify viable attached cells. Candidate cells were removed with a 30 µm capillary aspirating 0.1 µl per cell using automated bottom detection and scrape mode. Each removed cell was deposited into 10 µl of nuclease-free water supplemented with 1 µg/µl BSA (ThermoFisher) in a 384-well plate that was cooled during the cell picking. The plate was then sealed and placed on dry ice until use. Images were collected before and after picking, and only those samples that contained a single cell were used for further analysis. Negative controls were medium only. Half of the cell lysate (5 µl) was used in 20 µl ddPCR following the manufacturer's recommendations using ddPCR Mastermix for probes (BioRad) supplemented with 450 nM primers, 250 nM probes: *tRNA* 8A FAM and 9A HEX (Sigma Aldrich, sequences are in *Appendix 1—table 1* below). Annealing temperature of 57°C was used for the PCR run in a C1000 thermocycler (BioRad). Droplet analysis was performed using the QX200 instrument (BioRad), and data were analysed with Quantasoft according to the digital MIQE guidelines (*Huggett et al., 2013*).

**Appendix 1—table 1.** Primer and probe sequences used in sc/ddPCR. *Capital letters indicate LNA bases; polymorphism is shown in gray.

| m_tRNA_Fw | gtcacaattctatctctagg |
| --- | --- |
| m_tRNA_Rv | ggttgaagaaggtagatg |
| m_tRNA_8A_FAM (host mouse) | taattagtTtAaAaAaAaTtAaTgattt |
| m_tRNA_9A_HEX (parental B16) | taattagtTtAaAaAaAaAtTaAtgattt |

*Polymorphism of the mtDNA *tRNA*[Arg] locus of C57 mice and B16 cells has been published (*Bayona-Bafaluy et al., 2003*; *Tan et al., 2015*).

The annealing temperature for the *tRNA* ddPCR assay was optimized using a gradient run (data not shown), and a simplified validation of the sensitivity of the *tRNA* assay was performed. To each 20 µl ddPCR reaction mix about 20,000 copies of parental mtDNA were added (≈1 ng of purified B16 DNA) and spiked with 1000, 100, 10 and 1 copy of host mtDNA. Loss of linearity was observed at a prevalence of 0.5% and 0.05% spiked in host DNA. Pure parental mtDNA gave a background corresponding to 0.18% of host mtDNA. This very low background is caused by cross reaction due to residual affinity of the 8A host probe to the 9A parental locus. The reciprocal cross reaction of 9A probe binding to 8A parental locus was not observed (*Figure 1A* and *Appendix 1—figure 1* for validation of sc/ddPCR). No template controls (NTC) as well as B16ρ[0] cells were negative. A single B16 cell contains about 100 copies of mtDNA (not shown); hence our ddPCR sensitivity of 0.5% is sufficient to detect a single host mtDNA molecule.

### Mitochondrial biochemistry and bioenergetics

The activity of SQR, SDH and CS were assessed as described (*Tan et al., 2015*). Lactate production, ATP levels and glucose uptake were evaluated by standard procedures (*Tan et al., 2015*).

## Real-time (RT) PCR

Real time PCR was applied to assess the level of mtDNA, real time RT-PCR was used for evaluation of expression of selected transcripts. Details as well as the list of primers were published earlier (*Tan et al., 2015*).

## Mitochondrial chromatin immunoprecipitation (mitoChIP) assay

Cells were crosslinked with 1% formaldehyde, harvested and cell pellets frozen. Pellets were then resuspended in the ChIP lysis buffer, sonicated for 12 min at cycles consisting of 15 s sonication at amplitude 5 (QsonicaxxX) and 30 s cooling in ice-water bath. Sheared chromatin was treated with RNAseA and proteinaseK, and subjected to agarose electrophoresis to confirm correct shearing. Dynabeads were pre-absorbed with salmon sperm DNA (0.2 mg/ml) and BSA (0.5%) overnight, and 20 µl of bead slurry was combined with 5 µg of sheared chromatin and 1 µl of either anti-POLG1 IgG (Abcam, ab128899) or non-specific rabbit IgG, and incubated overnight on a rotating platform. After extensive washing, chromatin was eluted and treated with RNAseA and proteinaseK, followed by purification using PCR clean-up columns (MachereyNagel). qRT-PCR was then run using the Eva Green system (Solis Biodyne) at 95°C for 12 min, 40 cycles of 95°C for 15 s, 60°C for 20 s, 72°C for 35 s, followed by melt curve analysis using primers for mouse POLG1: forward - TGA TCA ATT CTA GTA GTT CCC AAA A, reverse - ACC TCT AAT TAA TTA TAA GGC CAG G. Data were quantified against a non-specific IgG via the ΔΔCt method.

## Electrophoresis and western blotting

Cell as well as tumor lysates were prepared and electrophoresis/WB carried out by standard procedures.

## Confocal microscopy, STED super-resolution microscopy and transmission electron microscopy (TEM)

For live cell imaging of mitochondria and mtDNA, cells were grown in chamber slides (LabTek) and incubated for 30 min with 100 nM MitoTracker Red (Molecular Probes) and 5 µg/ml Hoechst 33342, and, subsequently for 4 min with 2 µg/ml EtBr. Images were captured and analysed using the FV1000 confocal microscope and Fluoview software (Olympus).

STED microscopy was carried out as follows. Cells were grown on high precision cover-slips over-night, fixed with 4% paraformaldehyde in PBS for 10 min at 37°C, permeabilised with the washing buffer (0.1 M glycin, 0.05% TWEEN, 0.05% Triton X-100, PBS), blocked with 5% normal goat serum (Vector Laboratories), and incubated with primary antibodies against DNA (PROGEN), TFAM (Abcam) or Tom20 (Santa Cruz Biotechnology). This was followed by appropriate secondary antibodies including Alexa Fluor555 goat anti-mouse IgM and Alexa Fluor488 goat anti-rabbit IgG (H+L) (Life Technologies). Nuclei were labelled with Hoechst 33342 (Sigma-Aldrich). The cover-slips were mounted on glass slides using 90% glycerol mounting medium with 5% n-propylgalate. The samples were visualized using Leica TCS SP8 STED 3X microscope equipped with 660 nm STED depletion laser and fitted with the LAS X 64 bit package software with LAS AF SP8 Dye Finder, 3D visualization, deconvolution and co-localization module. The acquired images were

deconvolved by Huygens Professional software (Scientific Volume Imaging) and processed by FiJi ImageJ software.

TEM was performed using fixed cells via a standard procedure.

## Generation of tumours in C57BL/6N$^{su9\text{-}DsRed2}$ transgenic mice, cell sorting and confocal microscopy

B16$\rho^0$ cells with BFP nuclei were suspended in 100 µl PBS and grafted subcutaneously in C57BL/6N$^{su9\text{-}DsRed2}$ transgenic mice at $10^6$ cells per animal. The experiment was repeated 4 times, each time with five mice, and cells were recovered 4, 7, 9, 11 or 12 days post implantation. The pre-tumor lesion was resected and digested enzymatically (collagenase/DNase, 20 min, 37°C). The isolated tumour cells were sorted for nBFP- and mitochondrial DsRed-positive cells using the BD Influx high speed cell sorter (BD Biosciences) in 8-chambered cover-glass system (In Vitro Scientific). 12 hr after sorting, the cells were viewed using the Leica TCS SP5 AOBS Tandem confocal microscope equipped with the LAS AF software. The acquired images were deconvolved using the Huygens Professional software (Scientific Volume Imaging) and processed by FiJi ImageJ software.

## Respiration assays

Experiments evaluating intact cell respiration were performed using the Oxygraph-2k instrument (Oroboros) with cells suspended in RPMI medium without serum. Oxygen consumption was evaluated for cellular ROUTINE respiration, oligomycin-inhibited LEAK respiration, FCCP-stimulated uncoupled respiration capacity (ETS) and rotenone/antimycin-inhibited residual respiration (ROX). Respiration via mitochondrial complexes was evaluated using digitonin-permeabilised cells suspended in the mitochondrial respiration medium MiR06, and oxygen consumption was evaluated for CI-linked respiration, (CI+CII)-linked respiration, maximum uncoupled respiration, CII-linked uncoupled respiration as well as residual oxygen consumption. Respiration via CI and CII was evaluated in the presence of the proper substrates and inhibitors of the other complex. The results were normalized to the number of cells.

Respiration of tumour tissue was assessed basically as above. Following sacrifice of mice, tumours and livers (used as internal control) were excised and homogenized using the SG3 Shredder (Oroboros). The final cell suspension in the chamber contained 2 to 3 mg/ml of wet tissue. Mitochondrial respiration via CI or CII was evaluated as mentioned above.

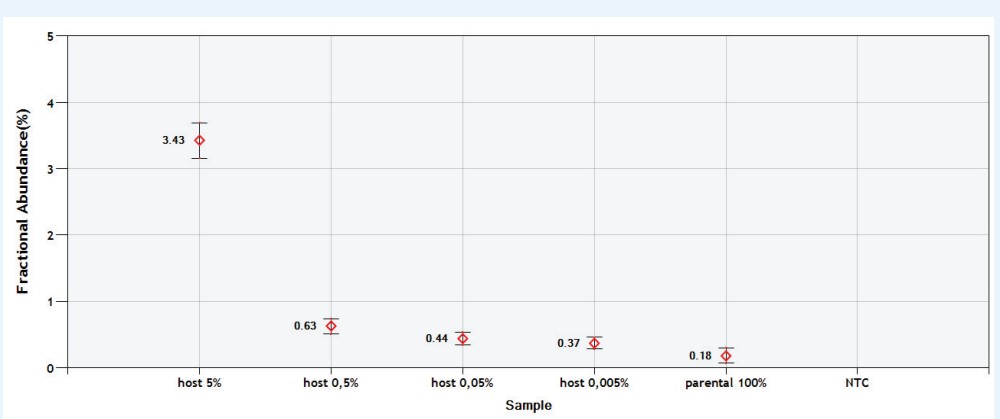

**Appendix 1—figure 1.** Validation of the sc/ddPCR assay for the *tRNA* locus. Parental mtDNA (about 20,000 copies) was spiked with 1000; 100; 10 and 1 copy of host mtDNA. Respectively, this corresponds to the expected fractions of 5; 0.5; 0.05% and 0.005%. Red diamonds indicate the measured fractions. NTC is non-template control.

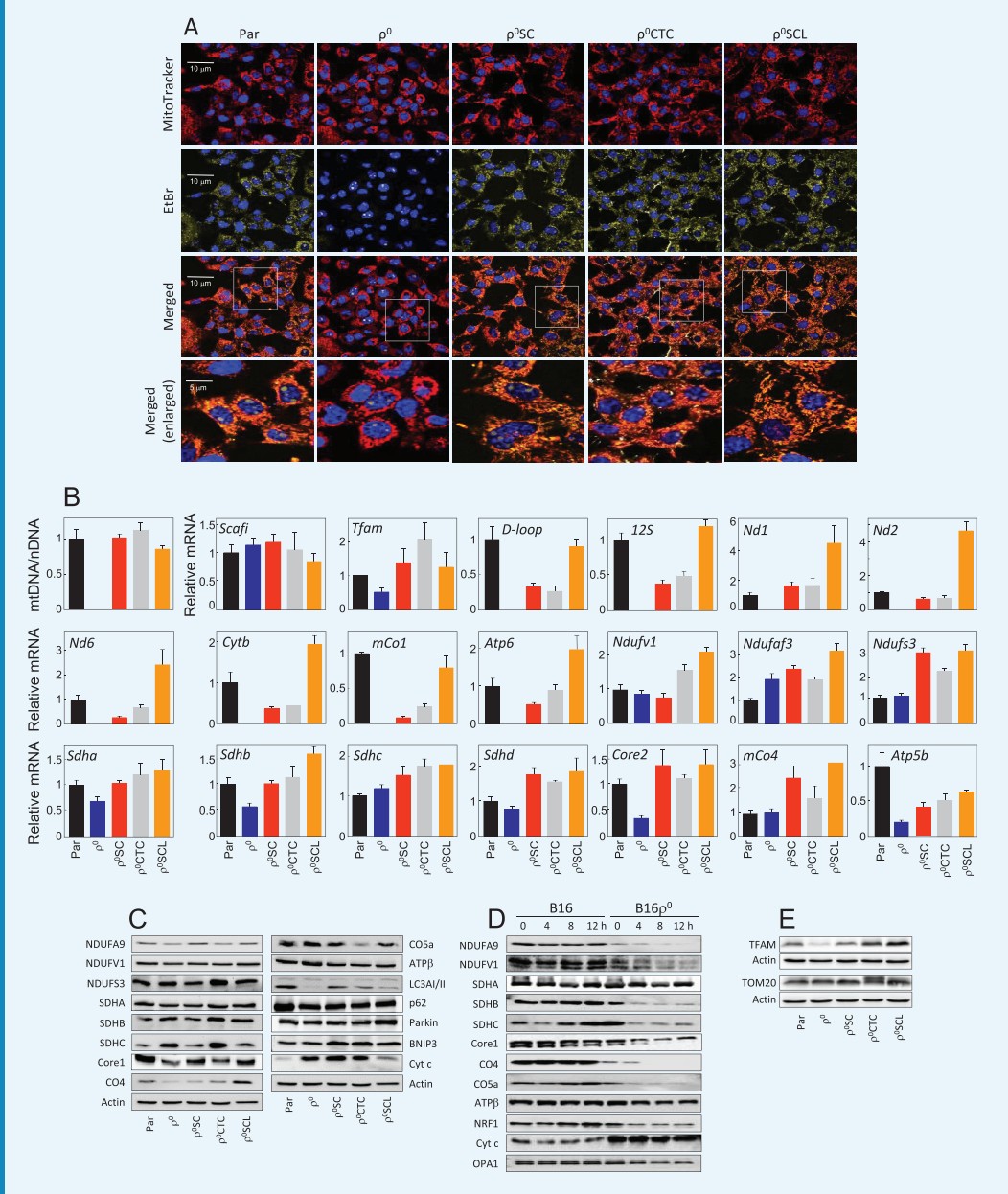

**Appendix 1—figure 2.** Tumour cells derived from primary tumour have restored mitochondrial function. (**A**) B16, B16$\rho^0$, B16$\rho^0$SC, B16$\rho^0$CTC and B16$\rho^0$SCL cells were stained with MitoTracker Red for mitochondria, EtBr for mtDNA and Hoechst 33342 for nuclei. (**B**) B16 sub-lines were probed using real-time (RT)PCR for expression of mtDNA and selected genes encoded by nDNA and mtDNA. The data are mean values ± S.D. derived from three individual experiments. (**C**) B16 sub-lines were subjected to SDS-PAGE followed by WB using various IgGs. (**D**) Control cells and cycloheximide-treated cells (5 µM, time as shown in the figure) were separated by SDS-PAGE and subjected to IgGs for selected mitochondrial proteins. (**E**) B16 sublines were subjected to SDS-PAGE followed by WB using anti-TFAM and anti-Tom20 IgG. Data shown are representative of three independent experiments.

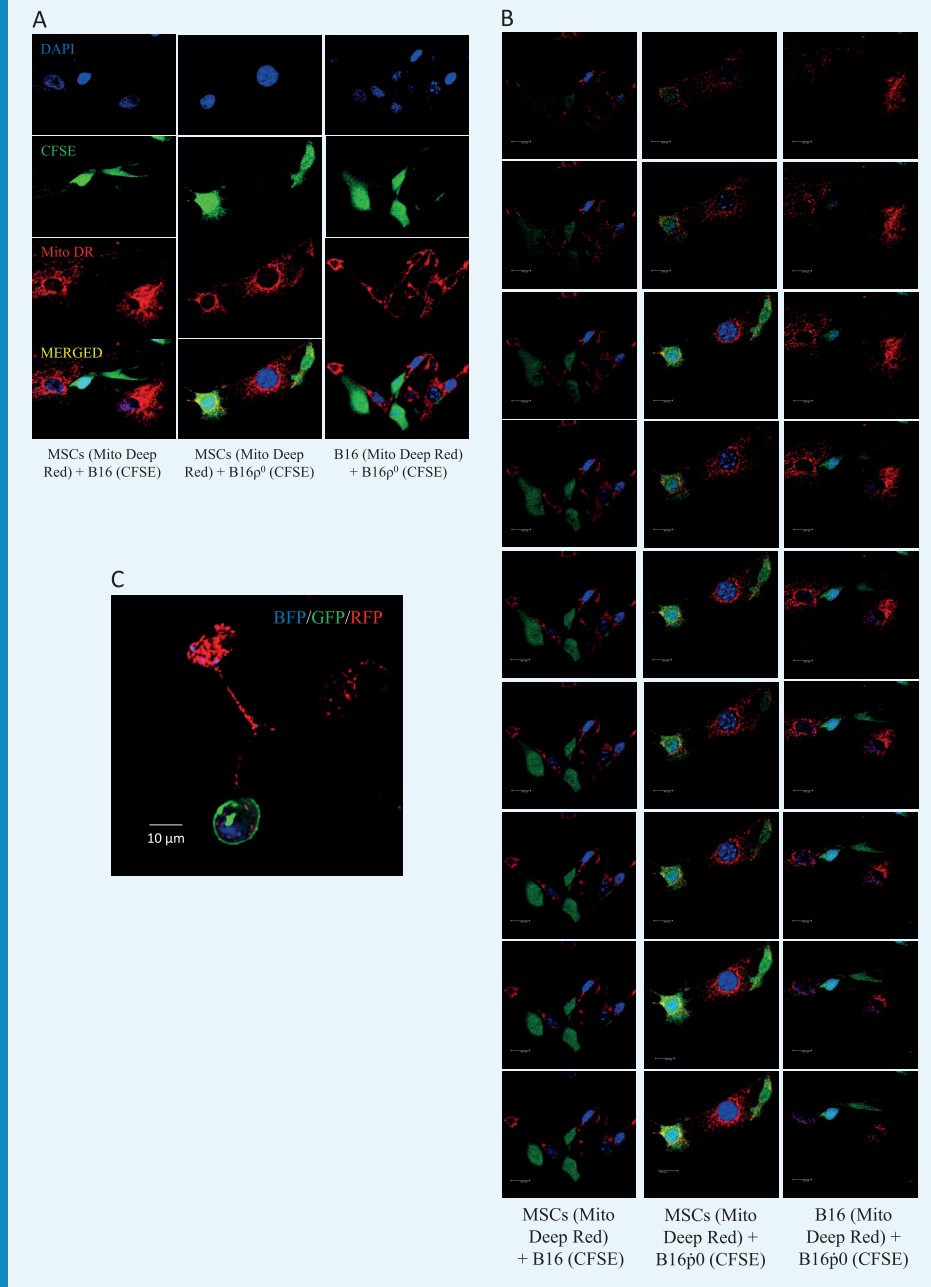

**Appendix 1—figure 3.** The efficiency of mitochondrial transfer during pair-wise co-culture of MSC, B16 and B16$\rho^0$ was assessed by confocal imaging after 24 hr co-culture in vitro. The donor mitochondria were labelled with MitoTracker Deep Red, a mitochondria targeted stain (red), while putative recipient cells were stained with cytoplasmic stain, CFSE (green), and the nucleus was stained with DAPI (blue). Cells were stained 2 hr prior to co-culture. Panel **A** shows representative images with separate colour channels for each co-culture pair, as labelled. Panel **B** shows Z stacks for the composite images. Panel **C** documents mitochondrial transfer from MSCs isolated from C57BL/6N$^{su9\text{-}DsRed2}$ mice with DsRed mitochondria co-cultured with $\rho^0$ cells labelled with nBFP and pmGFP.

