## [Decision Letter]

Thank you for submitting your article "Horizontal transfer of whole mitochondria restores tumorigenic potential in mitochondrial DNA-deficient cancer cells" for consideration by *eLife*. Your article has been favorably evaluated by Sean Morrison (Senior Editor) and three reviewers, one of whom, Ralph DeBerardinis (Reviewer #1), is a member of our Board of Reviewing Editors. The following individual involved in review of your submission has agreed to reveal their identity: Konstantin Khrapko (Reviewer #3).

The reviewers have discussed the reviews with one another and the Reviewing Editor has drafted this decision to help you prepare a revised submission.

Summary:

This paper examines the role of mitochondrial function in the growth of B16-derived melanoma xenografts in a syngeneic model. It follows from a previous study from the same authors demonstrating that rho0 B16 cells acquire mtDNA from the host during tumorigenesis. Here the authors provide two new types of information. First, they significantly extend their previous work to demonstrate the importance of, and selection for, cells with electron transport chain function during tumorigenesis in this model. Second, the authors use additional mouse models to provide evidence for direct transfer of intact mitochondria from the host into rho0 melanoma cells during tumor growth, as opposed to cellular fusion events or transfer of mtDNA. This is a provocative finding, novel to models of cancer biology.

Essential revisions:

1) In Figure 3, additional controls are needed. NDUFV1 and SDHC should be silenced with multiple independent shRNAs, ideally with rescue using cDNAs resistant to shRNA knockdown. Statistics are required for Figure 3.

2) Figure 4: It is unclear that the data represent true transfer of intact mitochondria. The MitoDR dye diffuses into the cell and then accumulates in the mitochondria. Therefore, simple dye transfer from donor to recipient would produce the same result. A better experiment would be to fuse GFP or another fluorescent marker to a nuclear-encoded protein that is targeted to the mitochondria.

3) Figure 4: It is unclear why mitochondria from MSCs are transferred to B16p˚ cells but not B16 cells. Because the analysis was performed just 12 hours after co-culture, the result should not be influenced by selective pressure. This implies a regulated process of transfer, e.g. structural changes that facilitate acceptance of transfer or a signaling-mediated event. If true, this would be the most interesting and surprising component of the paper. This should be studied more thoroughly.

4) Figure 4: More systematic analysis of the images, including analysis of z-stacks, is required to assure that the transferred mitochondria are really inside the recipient cells. In addition, it is unclear how the quantification of this result was done; more detail should be added to the legend. The middle image should have separate color channels.

5) Figure 4: It appears that the recipient cells contain many mitochondria, but to better understand the result, it is important to visualize all mitochondria in the double positive cells to determine the portion that are labeled with DsRed. Also, if mitochondria are transferred over the 11-day period of the tumor experiment, the labeled mitochondria would be expected to fuse with the rho-zero mitochondria, resulting in diffusion of the DsRed. This figure also needs data from several mice to assess the reproducibility of the findings and the frequency of double-positive cells, which appears to be extremely low.

6) Figure 4 is hard to interpret. It does not look like images of mitochondria and nucleoids from other studies. It would be helpful to outline the cell boundaries to help the reader understand the image.

---

## [Author Response]

*Essential revisions:*

1) In Figure 3, additional controls are needed. NDUFV1 and SDHC should be silenced with multiple independent shRNAs, ideally with rescue using cDNAs resistant to shRNA knockdown. Statistics are required for Figure 3.

The reviewers are correct that several different shRNAs should be used. In response to this well taken point, we state that we used four different shRNAs for both NDUFV1 and SDHC. We assessed the level of expression of the relevant proteins and respiration in cell lines transfected with all four shRNAs. In Figure 3, we document protein levels for shRNA#1 and shRNA#2 for both NDUFV1 and SDHC. In Figure 3 we only show proliferation and respiration (both routine and CI- and CII-dependent) for the more efficient shRNA#2 (for both NDUFV1 and SDHC). Since we then wanted to perform the experiments (respiration and tumour growth) using the shRNA with higher level of knock down, we focused on shRNAs #2 for both NDUFV1 and SDHC. We decided not to carry out the functional experiments with shRNAs #1 due to their less profound effect on the protein levels. The fact that in case of SDHC shRNA and, in particular NDUFV1 shRNA, we observed highly delayed tumour growth initiation or no tumour growth within 100 days after grafting the cells (with the corresponding ρ^0^ cells giving rise of tumour growth within 20-25 days), we believe that the knock down (also given its effect on respiration shown in Figure 3), is causally linked to knocking down of SDHC and NDUFV1, respectively. It is important that both SDHC and NDUFV1 knockdowns led to a delay in tumor formation. It is quite unlikely that both an SDHC and NDUFV1 shRNA would manifest an off-target effect that would result in a delay of tumor growth, it is much more likely that reduction in CI and CII activity induced by respective shRNA is at the bases of the observed phenomenon. This brings consistency to the reported results, as these are in fact two independent shRNAs directed at different components of the electron transport chain. Furthermore, to perform the in vivo experiments again with an independent set of shRNA would require at least 5 months, including all preparatory work, renewal of mouse work permits and the extended 100 days observation period in mice (cf. Figure 3). For these reasons, we asked the editors to consider exempting this point from the revision process.

In support of our reasoning re the point pertaining to the use of several different shRNAs, we discussed this point with the Editor of *eLife* and obtained the response as shown below, supporting our explanation above.

‘Thanks for your recent query that relates to point 1 in the reviewers' comments. I've shared your comments with the editors and they have the following comments:

The key experiment will be the one using the mitochondrially-targeted fluorescent protein to better substantiate the claim about horizontal transfer. It would be helpful if they referred to this exchange in their rebuttal letter, so that I remember that we decided to go forward without the additional shRNAs in the in vivo experiment.

As the editors point out, please refer to the above exchange so the editors remember what they decided to go with.’

We agree with the reviewers that Figure 3 requires statistical evaluation, and we performed it. This is now included in the revised version of the manuscript.

*2) Figure 4: It is unclear that the data represent true transfer of intact mitochondria. The MitoDR dye diffuses into the cell and then accumulates in the mitochondria. Therefore, simple dye transfer from donor to recipient would produce the same result. A better experiment would be to fuse GFP or another fluorescent marker to a nuclear-encoded protein that is targeted to the mitochondria.*

While in general we agree with reviewer’s concern in the matter of die leakage, we find it unlikely that this is an issue in our co-cultures of ρ^0^ cells with MitoDeepRed-labelled MSCs prepared from C57BLmice. The reason is that we do not see any mitochondria staining of parental cells co-cultured with MitoDeepRed-labelled MSCs, whereas if die leakage occurred from MCSs, the parental cell mitochondria would be stained as well, all the more so that these cells have higher mitochondrial membrane potential compared to ρ^0^ cells and would accumulate the die more readily. Notwithstanding this reasoning, we performed an additional experiment to unequivocally document intercellular transfer of whole mitochondria in MCSs co-cultures using nuclear-encoded mitochondria-targeted fluorescent protein as suggested by the reviewer. We used transgenic C57BL/6N*^su9DsRed2^* mice with red fluorescent mitochondria in somatic cells, from which we isolated MSCs and co-cultured them with ρ^0^ cells, whose nuclei were labelled with blue fluorescent protein (BFP) and plasma membrane with green fluorescence protein (GFP). Images in Appendix 1—figure 3C of the revised version of the manuscript document that mitochondria labelled with DsRed protein moved from MSCs to B16ρ^0^ cells. This confirms results in Figure 4, strongly supporting the case for mitochondrial transfer. We added a few sentences concerning this result in the Results section and also added legend to the new Appendix 1—figure 3C in the revised version of the manuscript.

*3) Figure 4: It is unclear why mitochondria from MSCs are transferred to B16p˚ cells but not B16 cells. Because the analysis was performed just 12 hours after co-culture, the result should not be influenced by selective pressure. This implies a regulated process of transfer, e.g. structural changes that facilitate acceptance of transfer or a signaling-mediated event. If true, this would be the most interesting and surprising component of the paper. This should be studied more thoroughly.*

We agree with the reviewers that this is a very intriguing point. In fact, we have started a separate project that will answer the very question why transfer of mitochondria from MSCs occursinto B16ρ^0^ cells but not into their normal counterparts. The group of our collaborator and co-author of this manuscript Prof. Agrawal published a paper recently showing that mitochondria move from MSCs to rotenone-stressed lung alveolar epithelial cells (Ahmad et al. Miro1 regulates intercellular mitochondrial transport & enhances mesenchymal stem cell rescue efficacy. EMBO J 33, 994-1010). We have now carried out similar experiments with MSCs isolated from C57BL mice and with parental B16 cells stressed with rotenone. As seen from Figure 8, there was over 3-fold increase in movement of mitochondria from MSCs into rotenone-treated B16 cells compared to their control counterparts, recapitulating our observation with B16ρ^0^ cells. We do not include these results in the revised version of the manuscript, since they are a part of another project (as eluded to above), in which we study the reasons for selective movement of mitochondria to cells lacking mtDNA or whose mtDNA is compromised in some way. The fact that we do see movement of mitochondria into B16 cells with normal mtDNA that were stressed with rotenone indicates that cells with compromised mitochondrial function (rotenone targets mitochondrial complex I) will import mitochondria in order to correct the situation, such as recover their respiratory capacity as we show in our manuscript. The nature of the signal that directs mitochondrial transfer to mitochondria-compromised cell is a matter of intense research in our laboratory.

Author response image 1.CFSE labeled B16 cells were co-cultured with mitoDR labeled MSC for 24 h.In identical experiments the B16 cells were pretreated with rotenone to induce mitochondrial dysfunction. Double positive cells were counted by FACS and also visualized on microscopy. Rotenone pre-treatment of B16 led to greater mitochondrial transfer and increase in double positive cells from about 2% to 7%. The geometric mean intensity of double positive cells also in-creased suggesting greater transfer.**DOI:**
http://dx.doi.org/10.7554/eLife.22187.010

*4) Figure 4: More systematic analysis of the images, including analysis of z-stacks, is required to assure that the transferred mitochondria are really inside the recipient cells. In addition, it is unclear how the quantification of this result was done; more detail should be added to the legend. The middle image should have separate color channels.*

In agreement with the request by the reviewers, we have now added the z-stack images for Figure 4 in the revised version of the manuscript as Appendix 1—figure 3B. As requested, we also included individual colour channels in the middle image of Figure 4 as Appendix 1—figure 3A in the revised version of the manuscript. Additionally, we have added more explanation about the quantification of the results in Figure 4 as requested by the reviewers.

*5) Figure 4: It appears that the recipient cells contain many mitochondria, but to better understand the result, it is important to visualize all mitochondria in the double positive cells to determine the portion that are labeled with DsRed. Also, if mitochondria are transferred over the 11-day period of the tumor experiment, the labeled mitochondria would be expected to fuse with the rho-zero mitochondria, resulting in diffusion of the DsRed. This figure also needs data from several mice to assess the reproducibility of the findings and the frequency of double-positive cells, which appears to be extremely low.*

We basically agree with the reviewers. Before answering this point, it may be good to recapitulate the experiment we performed in Figure 4, where we document the presence of dsRed-positive mitochondria donated by the stroma cells of the C57BL/6N*^su9-DsRed2^* mouse in the cytoplasm of the grafted, nuclear BFP-expressing B16ρ^0^ cells. The experiment itself is quite complicated. First, one needs to take into consideration that mitochondria move across over a relatively long period of time. We now have data from a different cell line from another project documenting that mitochondria start to move from the mouse stromal cells to grafted ρ^0^ cells within 5 days (will be published elsewhere). Also, using single cells real-time PCR, we observe continuous movement of mitochondria between stromal cells and grafted ρ^0^ cells for some 20 days following grafting of ρ^0^ cells, until all cells in the small tumour that starts to grow contain mitochondria with DNA. These new, unpublished data are from experiments with 4T1ρ^0^ cells, where the principle of mitochondrial transfer is similar to that of mitochondrial transfer to B16ρ^0^ cells. An important aspect of the experiment shown in Figure 4 stems from the fact that the MitoDsRed protein is coded for in the nucleus of the host (stromal, presumably mesenchymal stem) cells. Once the mitochondrion moves across to the grafted B16ρ^0^ cell, the MitoDsRed protein will ‘fade’, since it is not replenished – the recipient ρ^0^ cell does not code for this transgene. This in itself is a good evidence of selective mitochondria transfer as opposed to other possible mechanisms such as cell fusion. Therefore, at any point in time, we can visualise only a small percentage of mitochondria that would have moved across and would not have yet lost the red fluorescence emitted by the MitoDsRed protein.

We have performed the experiment in which we grafted B16ρ^0^ cells with BFL nuclei into C57BL/6N*^su9-DsRed2^* mice 4 times. In each experiment, we subjected the cells recovered from the animal after being sacrificed to fractionation by cell sorting 4, 7, 9, 11 or 12 days post-grafting. Prompted by the reviewer, we analysed data from available sorts and the average percentage of double-positive (DP) cells in the BFP-positive population was 0.23 ± 0.18, demonstrating that mitochondrial transfer is a reproducible phenomenon occurring in vivo at a relatively low, but stable frequency for an extended period of time. This frequency has now been included in the revised version of the manuscript.

We performed sorting of DP cells with very conservative settings, so that we were sure that we only recover DP cells without any contamination by single-positive cells or cell doublets. This stringency would cause exclusion of, for example, B16ρ^0^ cells with only a few red mitochondria and B16ρ^0^ cells were mitochondrial DsRed is already fading. Hence, the presented data give most likely some underestimation of the true in vivo mitochondrial transfer frequency.

Another, separate and more technical issue, is imaging of the double-positive cells that are recovered and sorted following subcutaneous grafting of B16ρ^0^nuBFP cells into C57BL/6N*^su9-DsRed2^* mice. As discussed above, we have to take into consideration the fact that, following transfer from stromal cells of the host into the grafted ρ^0^ cells, the MitoDsRed protein will fade rather fast, within about 20-24 h. Therefore, we only have a small window of opportunity to get an image of cells still expressing the mitochondrial red fluorescent protein. This involves first dissection of the tissue comprising the grafted cells, preparation of single cell suspension and sorting of DP cells with red and blue fluorescence. It takes up to 12 hours from sacrificing the animal to having cells sorted for further experimentation. For confocal microscopy, we need to seed the cells in special dishes and allow them to attach, at least to some extent, so that we can take reasonably clear images. We cannot allow the cells to be in the incubator for more than some 6-8 hours before imaging is performed, otherwise the red fluorescence will fade. Therefore, the images that we take in this way are not perfect, although they still clearly document red fluorescent mitochondria in cells with BFP nuclei. Later on, when we use these cells that are established as a subline, imaging is good but the cells do not comprise the MitoDsRed protein anymore. Therefore, we chose the best image we obtained. Additional images of B16ρ^0^DP cells recovered on different days following the engraftment of B16ρ^0^ nBFP cells into C57BL/6N*^su9DsRed2^* mice are presented in the attachment to this document, demonstrating that the images shown do not represent a one-off scenario. Given the technical difficulties of confocal imaging in this setup, the sorting data discussed above present a much more reliable evidence for the reproducibility of the mitochondria-transfer process in vivo.

The issue of potential fusion of original ρ^0^ mitochondria in B16ρ^0^ cells and mitochondria with mtDNA moving into these grafted cells form the host cells is of great interest. We do not know much about this process at this moment. In fact, we are researching this issue at the moment using the 4T1 cellular system and Balb-c mice in a separate project. We know that cells that contain both ρ^0^ mitochondria and mitochondria with mtDNA will not change their ratio much over longer periods in culture, which more-or-less excludes fusion of mtDNA-positive and mtDNA-negative mitochondria, which need to undergo fission in the course of cell division, at which stage they need to be shared relatively similarly between the daughter cells. A rather different issue is to resolve what happens under in vivo conditions, i.e. after cells with ρ^0^ mitochondria are grafted in mice and after they import mitochondria with mtDNA from the stroma. The original ρ^0^ mitochondria may fuse with the incoming mitochondria with mtDNA or ρ^0^ mitochondria may be removed by mitophagy, or both processes may occur in parallel. We know that at certain stage (about 20-25 days post grafting of 4T1ρ^0^ cells; unpublished data), the ρ^0^ mitochondria ‘disappear’ and the original ρ^0^ cells now contain virtually only mitochondria with mtDNA.

While we do not study in vivo fusion or fission of mitochondria in this manuscript (it is a separate and rather complex project), we subjected the B16 parental, B16ρ^0^ and B16ρ^0^DP cells to transmission electron microscopy. As seen in new Figure 4 in the revised version of the manuscript, B16 and B16ρ^0^DP cells comprise mitochondria with cristae, while B16ρ^0^ cells contain mitochondria with two concentric membranes with no cristae or with very low number of cristae. This suggests that defective mitochondria have been either cleared or reconstituted with mtDNA in vivo. Due to the new panel B in Figure 4, panels C, D, E and F of the original Figure 4 are panels D, E, F and G, respectively, in the revised version of the manuscript. We accordingly changed the Results section and legends to the figures. Also, we added a short chapter on TEM in the Supplementary Methods.

*6) Figure 4 is hard to interpret. It does not look like images of mitochondria and nucleoids from other studies. It would be helpful to outline the cell boundaries to help the reader understand the image.*

We apologise to the reviewer for omitting this in the original version. We now include in Figure 4 (and also in Figure 1) confocal images of whole cells with indicated region of interest (ROI) examined by STED microscopy. The higher magnification of STED unequivocally shows the presence of mitochondrial nucleoids within the examined ROIs in B16 and B16ρ^0^DP cells (and in B16ρ^0^SC, B16ρ^0^CTC and B16ρ^0^SCL cells), and their absence in B16ρ^0^ cells.